# Using machine learning-based variable selection to identify hydrate related components from FT-ICR MS spectra

Elise Lunde Gjelsvik[1]*, Martin Fossen[2], Anders Brunsvik[2], Kristin Tøndel[1]

1 Norwegian University of Life Sciences, Faculty of Science and Technology, Aas, Norway, 2 SINTEF AS, Trondheim, Norway

☯ These authors contributed equally to this work.

* elise.lunde.gjelsvik@nmbu.no

**Data Availability Statement:** All 1 files are available from the Zenodo database (doi: 10.5281/zenodo.6524710).

## Abstract

The blockages of pipelines caused by agglomeration of gas hydrates is a major flow assurance issue in the oil and gas industry. Some crude oils form gas hydrates that remain as transportable particles in a slurry. It is commonly believed that naturally occurring components in those crude oils alter the surface properties of gas hydrate particles when formed. The exact structure of the crude oil components responsible for this surface modification remains unknown. In this study, a successive accumulation and spiking of hydrate-active crude oil fractions was performed to increase the concentration of hydrate related compounds. Fourier Transform Ion Cyclotron Resonance Mass Spectrometry (FT-ICR MS) was then utilised to analyse extracted oil samples for each spiking generation. Machine learning-based variable selection was used on the FT-ICR MS spectra to identify the components related to hydrate formation. Among six different methods, Partial Least Squares Discriminant Analysis (PLS-DA) was selected as the best performing model and the 23 most important variables were determined. The FT-ICR MS mass spectra for each spiking level was compared to samples extracted before the successive accumulation, to identify changes in the composition. Principal Component Analysis (PCA) exhibited differences between the oils and spiking levels, indicating an accumulation of hydrate active components. Molecular formulas, double bond equivalents (DBE) and hydrogen-carbon (H/C) ratios were determined for each of the selected variables and evaluated. Some variables were identified as possibly asphaltenes and naphthenic acids which could be related to the positive wetting index (WI) for the oils.

## Introduction

One of the major flow assurance challenges in the oil and gas industry is the formation of gas hydrates and their agglomeration, causing complete blockage of pipelines [1]. Gas hydrates are formed under low temperatures and high pressures, as guest molecules are trapped inside and help stabilise crystalline cages consisting of water molecules held together by hydrogen bonds.

**Funding:** Funding from the Norwegian Research Council, Equinor ASA, OMV (Norge) AS, Wintershall DEA Norge AS and TotalEnergies. This work is a part of the Knowledge-Building Project for Industry (PETROMAKS 2), Project number: 294636 "New Hydrate Management: New understanding of hydrate phenomena in oil systems to enable safe operation within the hydrate zone".

**Competing interests:** The authors have declared that no competing interests exist.

Remediation methods consists of thermodynamic inhibitors (methanol, ethanol or glycols), low dosage hydrate inhibitors (LDHIs), or by ensuring operation outside the hydrate region by controlling the pressure and/or temperature [2]. However, operating outside the hydrate region is not always possible or economically feasible and chemicals have negative environmental impacts and should be avoided if possible. Previous experiments have shown that some crude oils form gas hydrates that do not agglomerate or deposit, but remain as transportable particles [3–5]. This can be explained by the existence of naturally occurring components in the crude oils with hydrate active properties that can interact with and alter the surface wetting properties of the hydrate particles from being hydrophilic to becoming hydrophobic, thus preventing agglomeration [6]. Despite a lot of research on the topic, the nature and structure of the hydrate active components in crude oils have not yet been determined in detail.

To prevent agglomeration of the hydrate particles, their wettability state must be controlled. Oil-wet particles are hydrophobic and associated with non-aggregating and thus flowable dispersions, while water-wet particles are hydrophilic and associated with aggregating hydrate particles with a higher potential for plugging [7]. The particles' wettability can be affected by the crude oil composition by adsorption or inclusion of components naturally occurring in crude oil to the hydrate surface.

Petroleum acids have shown surface activity towards hydrate surfaces. It has therefore been suggested that naturally occurring hydrate inhibiting components are present in the acid fractions of crude oils [8–11]. Furthermore, the acid fractions have been shown to contain large amounts of naphthenic acid compounds [12]. They consist of a complex mixture of alkyl-substituted acyclic and cycloaliphatic carboxylic acids with the general formula $C_nH_{2n+z}O_2$ where $n$ corresponds to the number of carbon atoms and $z$ specifies the hydrogen deficiency from ring formation [13]. Comparatively, asphaltene fractions are known to possess self-agglomerating properties and can stabilise oil-wetted systems [14]. It has been shown that the asphaltene fractions able to stabilise oil-wetted systems often are more polar, with higher oxygen content, higher acidity and lower DBEs [15]. Other studies have suggested that the possible hydrate activity of asphaltenes is related to their sulfoxide content [16]. Accordingly, some asphaltenes can alter the plugging potential of crude oils [17, 18].

The complex mixture and relatively high masses of the components in crude oils make it difficult to identify single components with most mass spectrometers. However, with the high mass accuracy of Fourier Transform Ion Cyclotron Resonance Mass Spectrometry (FT-ICR MS) more detailed analysis of crude oils with the ability to identify a large number of polar groups, including compounds present in low concentrations, is possible [19]. FT-ICR MS has previously been used extensively for crude oil characterisation [20–27]. Qian et al. [28, 29] showed that electrospray ionisation (ESI) FT-ICR MS was able to identify more than 3000 chemical formulas of nitrogen containing aromatic compounds in positive mode. Additionally, studies have shown that asphaltenes can be characterised by positive mode ESI FT-ICR MS [30–32].

With the highly detailed spectra derived from FT-ICR MS, there is a need for powerful data analysis methods to efficiently extract valuable information and disregard unimportant information. The present work describes the use of machine learning-based variable selection for the identification of naturally occurring hydrate inhibitors from ESI positive FT-ICR MS spectra and relating the selected variables to the wettability state of the respective crude oils.

## Materials and methods

### Fluid system

The crude oils used originated from the Norwegian continental shelf and were used as received unless specifically mentioned. The water phase consisted of 3.5 wt% NaCl in tap water, thus

containing only monovalent ions in the water-phase, which simplifies the water chemistry avioding possible unwanted reactions by bivalent ions such as $Ca^{2+}$ [33]. The gas phase was a mixture of 86/8/6 mol% of methane, ethane and propane respectively (Linde Gas AS) with a mixture tolerance of 10% and an analysis uncertainty of 2%.

## Experimental set-up

The autoclave used in the experiments was a 200 mL high-pressure sapphire cell (Top Industrie) owned by SINTEF AS, placed inside a temperature controlled chamber. The temperature was measured using a PT-100 element positioned at the bottom of the cell. A connected stirrer mixed the phases to create a fully dispersed system. The cell was fitted with a Hy-Lok FT Micron Tee Filter with a 150 $\mu$m sintered stainless steel filter element. A probe inserted from the top was used to measure the conductivity in the liquid phase. Gas filling was controlled using an IN-FLOW HI-Press MFC mass flow controller (Bronkhorst).

## Successive accumulation of hydrate active components

A successive accumulation procedure (spiking) was performed with the aim of accumulating possible hydrate active components. A schematic illustration of the procedure is shown in Fig 1. The method developed by Fossen et al. [34] was based on Borgund et al. [6] which presented the same procedure, but with a non-pressurised system using tetrahydrofuran as hydrate former. The procedure started with a fresh oil sample which was added to the cell with the water phase at a given water cut and pressurised with a hydrocarbon gas phase. The pressure used for the current study was 65 bar. The water cut is the ratio of water compared to the total volume of the system. The temperature was lowered to 2°C while stirring the liquid to ensure a homogeneous dispersion. By cooling the system at high pressure, the hydrate formation region will eventually be reached, and given enough sub-cooling, the system will form hydrates. For the current tests, the system was kept at low temperature over night, to ensure hydrate formation. When hydrates had formed, and the reaction allowed to reach equilibrium,

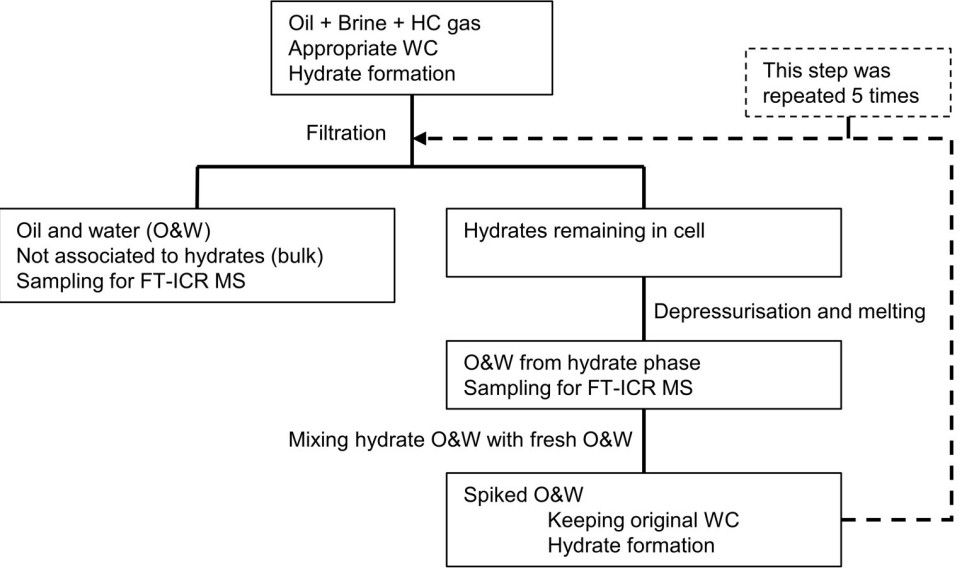

**Fig 1. Schematic illustration of the successive accumulation experiment for spiking of the hydrate phase.**

the phase not associated with hydrates, called the bulk phase, was drained through the bottom of the cell. The driving force for draining was the pressure difference of the cell and the ambient pressure conditions outside the cell. The hydrate phase was retained by the filter, so only water and oil not associated to hydrates were drained. Once the bulk phase was drained, the cell was depressurised and the temperature was increased, leading to dissociation of the hydrate phase which was drained and collected, resulting in an oil and a water phase that had been associated to the gas hydrates. The now liquid hydrate phase was then mixed with fresh oil and water at a ratio ensuring the same water cut as the previous run, before repeating the hydrate formation and draining procedure. Small samples were taken from both the bulk phase and the hydrate phase at each step for analysis by FT-ICR MS.

## Wetting index experiments

A wetting index (WI) procedure for determining the emulsion inversion point was developed by Høiland et al. [35] and advanced by Fossen et al. [34]. In short, the WI is obtained from determination of the inversion point of the emulsions with and without hydrates present. When the emulsion inversion point shifts towards higher water cuts after hydrate formation, the hydrates are oil-wetted, and when the shift is towards lower water cuts, the hydrates are water-wetted. This is in accordance with the principles of Bancroft [36]. The WI is defined as the normalised difference in inversion point with, and without hydrates present, represented by a number between -1 and +1. Positive values indicate oil-wetted systems with little or no potential of plugging, while negative values indicate water-wetted systems with a high potential of plugging. The absolute value of the WI number is expected to be of importance, and a higher positive or negative value indicates higher degrees of oil-wetted or water-wetted hydrate particles.

## FT-ICR MS analysis

For the FT-ICR MS analysis, the samples were prepared by dissolving 20 $\mu$L sample in 980 $\mu$L dichloromethane. 20 $\mu$L of the diluted sample was then added to 980 $\mu$L of a 1:1 mixture of toluene and methanol. 100 $\mu$L were injected onto the FT-ICR MS using a Aglient 1290 Infinity HPLC system as the introduction device. The 100 $\mu$L were injected over a period of 10 minutes with a flow of 10 $\mu$L per minute. The mass spectra were acquired using a Bruker Solarix XR FT-ICR MS (Bruker Daltonik GmbH, Germany) equipped with a 12 Tesla magnet (Bruker Biospin, France) owned by SINTEF and located in Trondheim (resolution: 450 000 at m/z 400). The FT-ICR was equipped with an electrospray ion source (ESI) operating in positive mode with the mass range set to 150–3000 m/z.

3 oil samples (anonymised to A, J2 and I) underwent the successive accumulation procedure resulting in 41 samples of different spiking levels. 6 spiking levels for oil A and 5 spiking levels for oil J2 and I. The samples were analysed by FT-ICR MS in three parallels each. For each sample, 220 spectra were collected.

## Data treatment

A bucket table was created of the data using Bruker Compass ProfileAnalysis 2.1. The settings in ProfileAnalysis was as follows: the average peak list was calculated, normalisation was set to the sum of bucket values in analysis, no baseline or smoothing, S/N threshold of 4, relative intensity threshold of 0.01 and absolute intensity threshold of 100. The resulting data set consisted of 123 samples and 27600 variables between m/z 148.44 and 1001.66.

## Principal Component Analysis (PCA)

PCA [37] is an unsupervised method for data reduction where a large data set $X$ is decomposed into a subspace containing linear combinations of the original variables as shown in Eq 1.

$$X = X_{In} wgt_X \tag{1}$$

Where $X_{In}$ has the shape $(N, K)$ and is the mass spectra for $N$ oil samples with $K$ $X$-variables and $X$ has the shape $(N, K)$ which are the balanced spectra for $N$ oil samples with $K$ variables. Eq 2 shows the PCA model for $A$ Principal Components (PCs).

$$X = \bar{x} + T_A P_A^T + E_A \tag{2}$$

Where $P_A$ are the loadings and orthonormal eigenvectors of $(X - \bar{x})^T (X - \bar{x})$ with shape $(K, A)$, minimising the covariance between the $X$-variables after $A$ PCs. The scores $(T_A)$ are orthogonal as shown by Eq 3 andwil have shape $(N, A)$.

$$T_A = (X - x) P_A \tag{3}$$

The error term in 2 is $E_A$ which is calculated by Eq 4.

$$E_A = X - \bar{x} - T_A P_A^T \tag{4}$$

## Variable selection methods

Variable selection is the process of selecting a subset of relevant variables to use when constructing a model. When a data set contains a large number of variables, it is often assumed that the data contains irrelevant or redundant variables that can be removed without loss of information. Removing them can improve the prediction ability of the model and reduce the computational cost during modelling. Variable selection can also be used to identify the features with the highest correlation to the response, i.e. the most important variables.

In this paper, variable selection methods such as Partial Least Squares Discriminant Analysis, Decision Trees, Random Forest, Boosting, and LASSO (Least Absolute Shrinkage and Selection Operator) regularisation were compared with the aim of predicting whether the samples were related to the hydrate or the bulk phase. An attempt was made to identify components in the data related to hydrate formation with the hypothesis that there could be systematic differences in the spectra which the proposed methods could be able to distinguish.

**Partial Least Squares Discriminant Analysis (PLS-DA).** PLS-DA [38] decomposes large data sets into a subspace of latent variables consisting of scores and loadings which represent the main features of covariance in the data. The latent variables are found by a maximisation of the covariance between the features, $X$ and the response, $Y$. $X$ has the same input model as for PCA, shown in Eq 1. As PLS-DA also takes the response into account as opposed to PCA, the input model for $Y$ is shown in Eq 5.

$$Y = Y_{In} wgt_Y \tag{5}$$

Where $Y_{In}$ has the shape $(N, J)$ and is the input categorical variables (0 or 1) for $N$ oil samples with $J$ categorical variables, $wgt_X$ are the statistical weights for balancing the sum of squares for the $Y$ variables and $Y$ is the balanced data with shape $(N, J)$ for $N$ oil samples with $J$ $Y$-variables. The decomposition of $X$ is taken into account, resulting in $Y$ relevant latent variables. This is shown by Eqs 6 and 7.

$$X = \bar{x} + T_A P_A^T + E_A \tag{6}$$

$$Y = \bar{y} + U_A Q_A^T + F_A \tag{7}$$

Where $A$ denotes the number of PCs used and $E_A$ and $F_A$ are the error terms using $A$ PCs. Plotting of these latent variables provides overview of co-variations both within and between model inputs and outputs. The loading weight matrix ($W_A$) maximises the covariance between $X$ and $Y$ by maximising the covariance between $T$ and $U$ after $A$ components. The scores ($T_A$) are orthogonal as shown by Eq 8.

$$T_A = (X - \bar{x}) \times W_A \tag{8}$$

The loadings for $X$ ($P_A$) are calculated by Eq 9 while the loadings for $Y$ ($Q_A$) are calculated by Eq 10.

$$P_A = (T_A^T T_A^T)^{-1} T_A^T (X - x) \tag{9}$$

$$Q_A = (T_A^T T_A^T)^{-1} T_A^T (Y - y) \tag{10}$$

The error term for $X$ ($E_A$) is calculated as for PCA in Eq 4 and the error term for $Y$ ($F_A$) is calculated by Eq 11.

$$F_A = Y - y T_A Q_A^T \tag{11}$$

The regression coefficients ($B_A$), which are measures of the impact of variations in the various features on the respective response variables, are calculated by Eq 12.

$$B_A = W_A Q_A^T \tag{12}$$

Prediction of $Y$ is then obtained by Eq 13 where $b_0$ is the intercept.

$$Y_{pred} = b_0 + X_{new} B_A + F_A \tag{13}$$

When $Y$ is categorical and the problem is classification, Linear Discriminant Analysis (LDA) is used to predict the class membership of the samples from the PLS-DA component construction by encoding the class membership of the observed variables in $X$ into 0 or 1 [39].

PLS-DA can be used for variable selection by calculation of the Variable Importance in Projection (VIP) for each $X$ variable in the PLS-DA model. The VIP score summarises the influence of the individual $X$ variables on the PLS-DA model and are calculated as the weighted sum of squares for the PLS-DA weights $w_j$ which takes the amount of explained variance in $Y$ into account for each extracted latent variable. VIP therefore gives a measure that can be used to select variables which contribute the most to the explanation of the variance in $Y$. The VIP score for variable $K$ can be calculated from Eq 14.

$$VIP_K = \sqrt{n \frac{\sum_{j=1}^{A} B_j^2 t_j^T t_j \left( \frac{w_{kj}}{\| w_j \|} \right)}{\sum_{j=1}^{A} B_j^2 t_j^T t_j}} \tag{14}$$

Where $B$ is the regression coefficient matrix, $w_j$ is the weight vector, $w_{kj}$ is the $k$th element of $w_j$ and $t_j$ the score vector from the PLS-DA model with $A$ PCs. A variable with a VIP score greater then 1 are generally considered as important, however this limit is sensitive to non-relevant information in $X$ [40]. In this study, the threshold for selecting variables were determined as the point where the VIP-values flatten out, which was found to be 5.

**Decision Trees (DTs).** DTs [41, 42] are models where decisions are made by asking a series of questions and generating decision rules based on them. These models consist of a tree root, decision nodes, branches and leaf nodes. They aim to find the smallest set of rules that is consistent with the training data. In general, the rules have the form: *if condition$_1$ and condition$_2$ and condition$_3$ then outcome* and are chosen to divide observations into segments that have the largest difference with respect to the target variable. Therefore, the rule selects both the variable and the best break point (usually selected by significance testing or reduction in variance criteria) for maximal separation of the resulting subgroups.

To avoid overfitting, the trees often have to be pruned by setting a limit for the maximal depth. A leaf can no longer be split when there are too few observations, the maximum depth (the hierarchy of the tree) has been reached, or no significant split can be identified. It is assumed that observations belonging to different classes have different values in at least one of their features. DTs are usually univariate, since they use splits based on a single feature at each internal node.

**Random forest (RF).** In DTs, the initial selected split effects the optimality of variables considered for subsequent splits, making these methods prone to overfitting and other problems. This can be handled by introducing RF [43–45], an ensemble tree method where each tree is based on a random subset of the data and its features (selected by bootstrapping). The advantage of ensemble trees is that the trees are grown with varying initial splits, and either a voting or the average of the predictions for each new data point across all trees is used. The vote distribution can be used to develop a non-parametric probabilistic predictive model. The change in prediction accuracy when the values of a feature are randomly permuted among the observations gives estimates of the importance of each feature.

**Ensemble learning.** Ensemble learning combines weak classification models with the main idea that many models in combination perform better than one model alone [46].

Boosting [47] is an ensemble learner where weak learners are trained sequentially, trying to improve upon its predecessor. The classifiers emphasise errors made by the previous classifier, aiming at decreasing the model bias. Boosting learners combine underfitting models with low prediction accuracy with the aim of improving the final prediction. Gradient Boosting [48, 49] is a boosting method where trees are built in every iteration, always minimising the prediction error of the classifier. This combination of several smaller trees forms a stronger learner able to fit larger parts of the data than a simple decision tree can. XGBoost (eXtreme Gradient Boosting) [50] is another boosting method based on gradient boosting, which introduces a penalty function in the boosting algorithm and utilise the computational power more efficiently, reducing the computation times.

**Regularisation.** Another type of machine learning method valuable for variable selection purposes is the regularisation-based method LASSO (least absolute shrinkage and selection operator) [51].

**LASSO.** In LASSO the estimates of the regression coefficients are obtained using L1-constrained least squares. This forces the sum of the absolute values of the regression coefficients to be less than a fixed value, which forces certain coefficients ($\beta_j$) to be set to zero. The variables which have their regression coefficients set to zero, are omitted from the model. LASSO minimises Eq 15 where the ordinary least squares (OLS) problem is the first term with $\beta_0$ as the intercept, and the second term $\lambda \sum_{i=1}^{p} |\beta_j|$ is the regularisation term.

$$RSS_{LASSO} = \sum_{i=1}^{n} (\boldsymbol{y}_i - \beta_0 - \sum_{j=1}^{p} \beta_j \boldsymbol{X}_{ij})^2 + \lambda \sum_{i=1}^{p} |\beta_j| \tag{15}$$

**Variable importance score.** For each of the variable selection methods a variable importance score can be computed, which is a measure of the variables' relative importance in the prediction model. These scores therefore reflect which variables are the most relevant for the target and which variables are of least importance.

The variable importance score can also be used to improve the prediction model by including only the variables with high scores in the model.

## Data analysis

All statistical methods were implemented using Python 3.8 and its machine learning packages. The response consisted of a vector containing information of the samples origin, either extracted from the hydrate phase or from the bulk phase. For the linear models, PCA, PLS-DA and LASSO, the data set was standardised (standard deviation = 1) and mean centered (mean = 0). For PLS-DA, the optimal number of components were selected by splitting the training set into two, 70% for calibration and 30% for validation, and finding the most commonly selected number of components by calculating the accuracy over 25 splits. All methods were validated using 25 different training and test set splits with 70% in the training set and 30% in the test set. Molecular formulas were determined using Bruker Compass DataAnalysis 5.0. From the peak corresponding to the m/z of the variables selected, the formula best fitting to the peak was chosen.

## Results

### Wetting index experiments

The three oils underwent the WI experiment and their WIs were calculated. The WI for oil A was shown to be 0, indicating that it has no clear plugging tendency. Oil J2 and I were determined to have positive WIs of 0.44 for oil J2 and 0.31 for oil I, indicating that they have low or no tendency of plugging. The resolution of the measurements in terms of water cut were 10 volume%. This gives an accuracy of the measurement of ±0.05 volume% and thus a corresponding uncertainty in the measured WIs. Evaluation of the sensitivity of the water cut resolution on the WIs was not performed in this study.

### PCA

Each of the oil samples were analysed by PCA and the resulting scoreplot of the first Principal Component (PC1) and the second Principal Component (PC2) for the data set is shown in Fig 2 where the samples are identified by the oil they originated from. The same scoreplot is shown in Fig 3 with the samples distinguishing the individual spiking levels. In both figures, the samples from the bulk phase are shown in the plot to the left and the samples from the hydrate phase are shown in the plot to the right. Fig 2 shows the differences between the three oils, and PC1 shows the difference between the samples from oil J2 and the samples from the two other oils, A and I. Additionally, PC1 shows a separation between the samples that have undergone the spiking experiment, and the crude oil samples which are clustered around 0. PC2 shows differences in the spiking samples from oil A and I. The spiking samples for oil J2 are clustered at 0 for PC2.

Fig 3 shows the differences between the spiking levels along PC2.

### Comparing mass spectra

To investigate differences between the spiking levels of the hydrate phase in each of the oils, the mass spectra from each spiking level were compared to a sample which had not been

PCA scoreplot

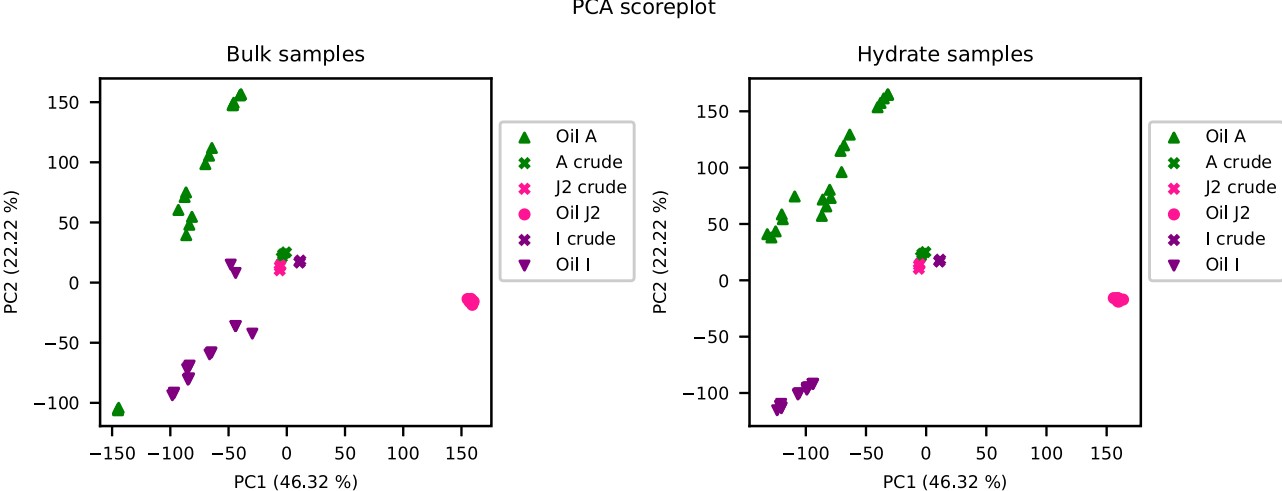

**Fig 2. PCA scoreplots with samples from the bulk phase shown in the left plot and samples from the hydrate phase shown in the right plot.**
Samples are coloured according to which crude oil they originated from and the crude oils have the symbol x.

spiked. This was only performed for the hydrate phase as it was assumed that the hydrate active components would be present in this phase. An average spectrum was calculated from the tree parallels for each spiking level and for samples removed before the spiking experiment. The sample removed before spiking is referred to as spiking level 0. The mass spectra for spiking level 0 was subtracted from the spectra for the remaining spiking levels for each of the oils. The results for oil A are shown in Fig 4. From Fig 4, four m/z values appeared to have an increasing trend as the spiking levels increased for oil A. They are shown in Table 1 with the molecular formula, double bond equivalent (DBE), the degree of unsaturation of the molecule, the hydrogen-carbon (H/C) ratios, which adduct the molecule has, either sodium (Na) or hydrogen ($H^+$) and the molecular weight.

PCA scoreplot

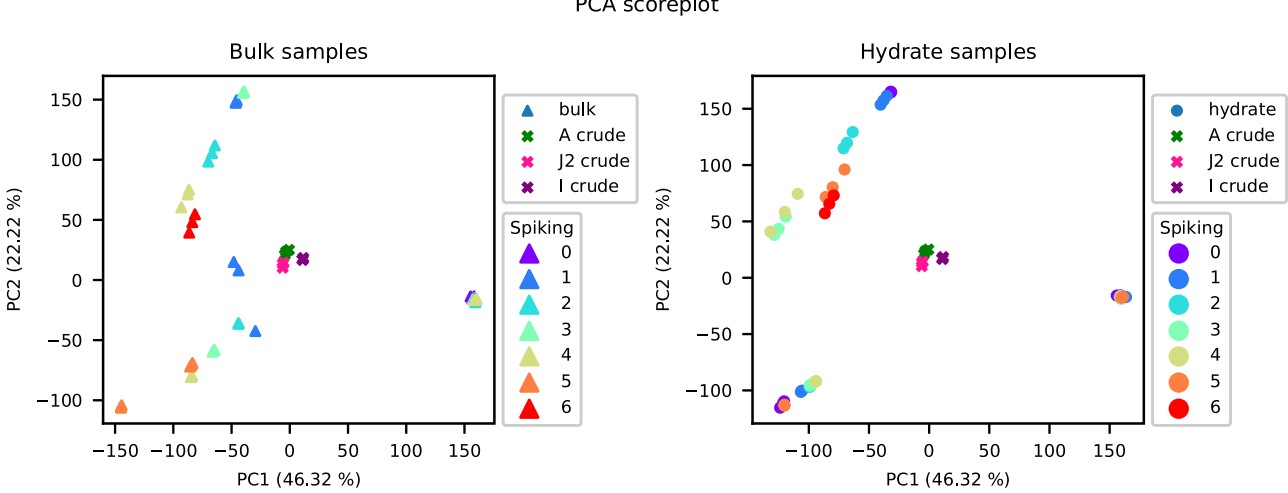

**Fig 3. PCA scoreplots with samples from the bulk phase shown in the left plot and samples from the hydrate phase shown in the right plot.**
Samples are coloured by spiking level and the crude oils have the symbol x.

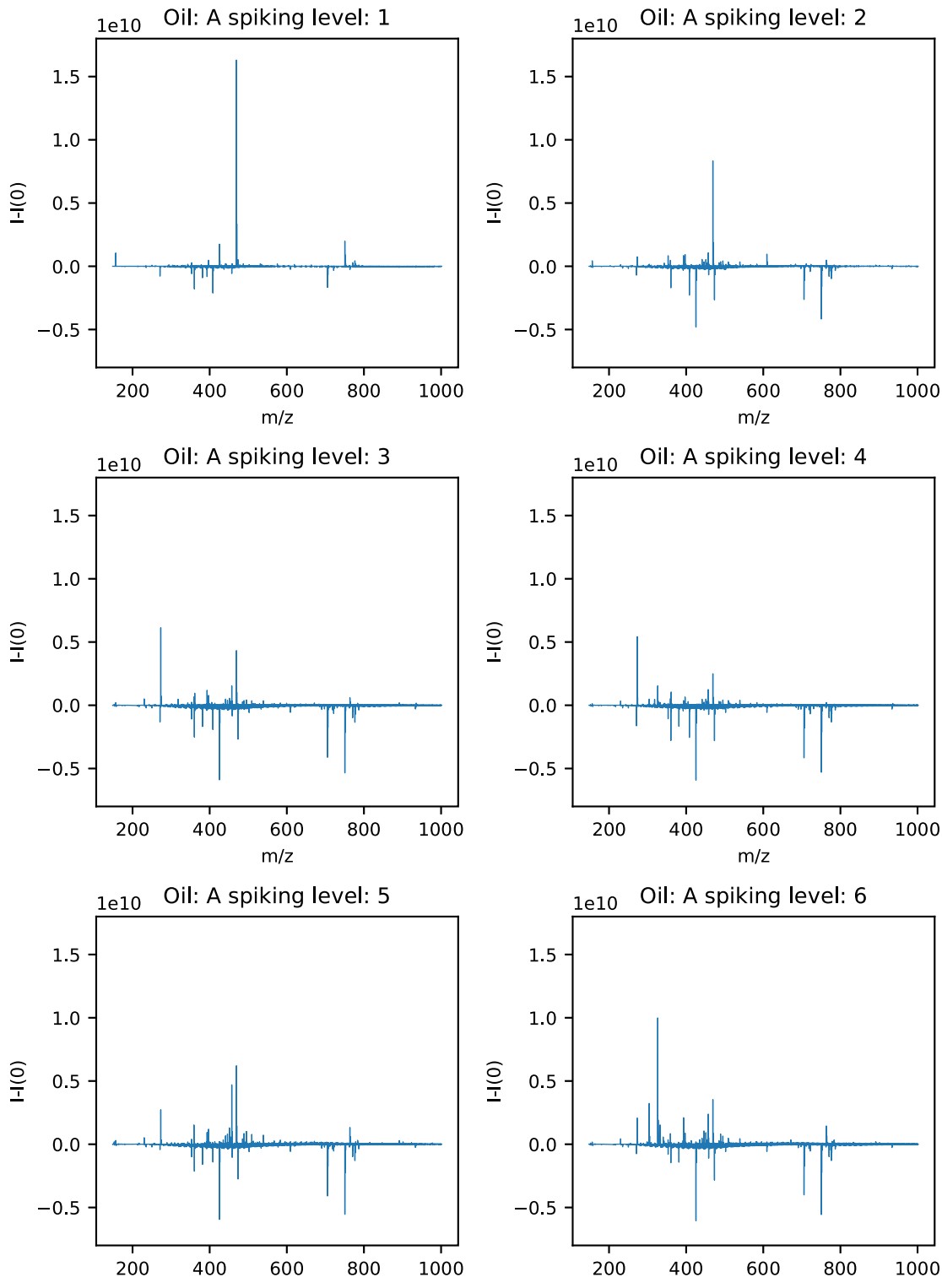

**Fig 4. Mass spectra of samples from the hydrate phase for oil A with spiking level 0 subtracted from each of the spiking levels 1–6.**

**Table 1. Peaks increasing for oil A.**

| m/z | Formula | DBE | H/C | Adduct | Molecular formula |
|---|---|---|---|---|---|
| 273.17 | $C_{12}H_{26}O_5$ | 0 | 2.17 | Na | 250.1780 |
| 397.18 | $C_{18}H_{30}O_8$ | 4 | 1.67 | Na | 374.1941 |
| 457.28 | $C_{22}H_{42}O_7$ | 2 | 1.91 | Na | 434.2880 |
| 469.31 | $C_{28}H_{46}O_4$ | 10 | 1.64 | Na | 446.3244 |

The m/z values with increasing trend as spiking levels increased for oil A, their molecular formula, DBE, H/C-ratio, which adduct the molecule has, Na or $H^+$, and the molecular weight

The results with spiking level 0 subtracted from the remaining spiking levels for oil J2 are shown in Fig 5. For oil J2 no distinct m/z values increased with increasing spiking levels.

The results with spiking level 0 subtracted from the remaining spiking levels for oil I are shown in Fig 6. From Fig 6, two m/z values appeared to have an increasing trend as the spiking level increased for oil I. They are shown in Table 2 with molecular formula, DBE, H/C-ratio, which adduct the molecule has and the molecular weight. Additionally, the variable m/z 156.44 increased, but this is an ion with charge three from the m/z 469.32 peak and is therefore not reported.

## Variable selection

Several variable selection methods such as Decision Trees, Random Forest, Gradient Boosting, XGBoost, LASSO regularisation and PLS-DA through VIP were tested with the aim of finding components related to hydrate formation. The samples were classified by their origin, whether they were sampled from the bulk phase (0) or from the hydrate phase (1). During the data analysis, it was discovered that the accuracy of the models depended on the composition of the training and test sets. This is an indication that the samples have such large variation between them that some compositions of the training set are not able to predict the test set. This was overcome by running 25 different training and test set combinations. The variable selection methods were tested on all variables to evaluate which method predicted the samples most accurately. The accuracy scores of the test set for each of the six methods are shown in Fig 7, where the accuracy is defined as the fraction of correctly classified samples. The distributions in accuracy for each method is shown by the bars in Fig 7. The best performing model was PLS-DA with an accuracy of 0.62 ± 0.12.

The performance for each of the variable selection methods is shown in Table 3. Each time a model was fitted to a new training and test set, the variables selected by the model were extracted. Variables that were selected by several different training/test sets are more likely to be related to hydrate formation. For the best performing variable selection method, PLS-DA, 26 variables were selected as important by all of the 25 models out of the total 27600 variables in the data set. However, during inspection of the m/z-values, it became apparent that two of the variables referred to the same peak. Additionally, two variables were the corresponding isotope peak, for m/z 393.30 (isotope peak: 394.30) and 469.32 (isotope peak: 470.32). The variables were combined, resulting in 23 unique selected variables which are shown in Table 4 with molecular formula, DBE, H/C-ratio, which adduct the molecule has and molecular weight.

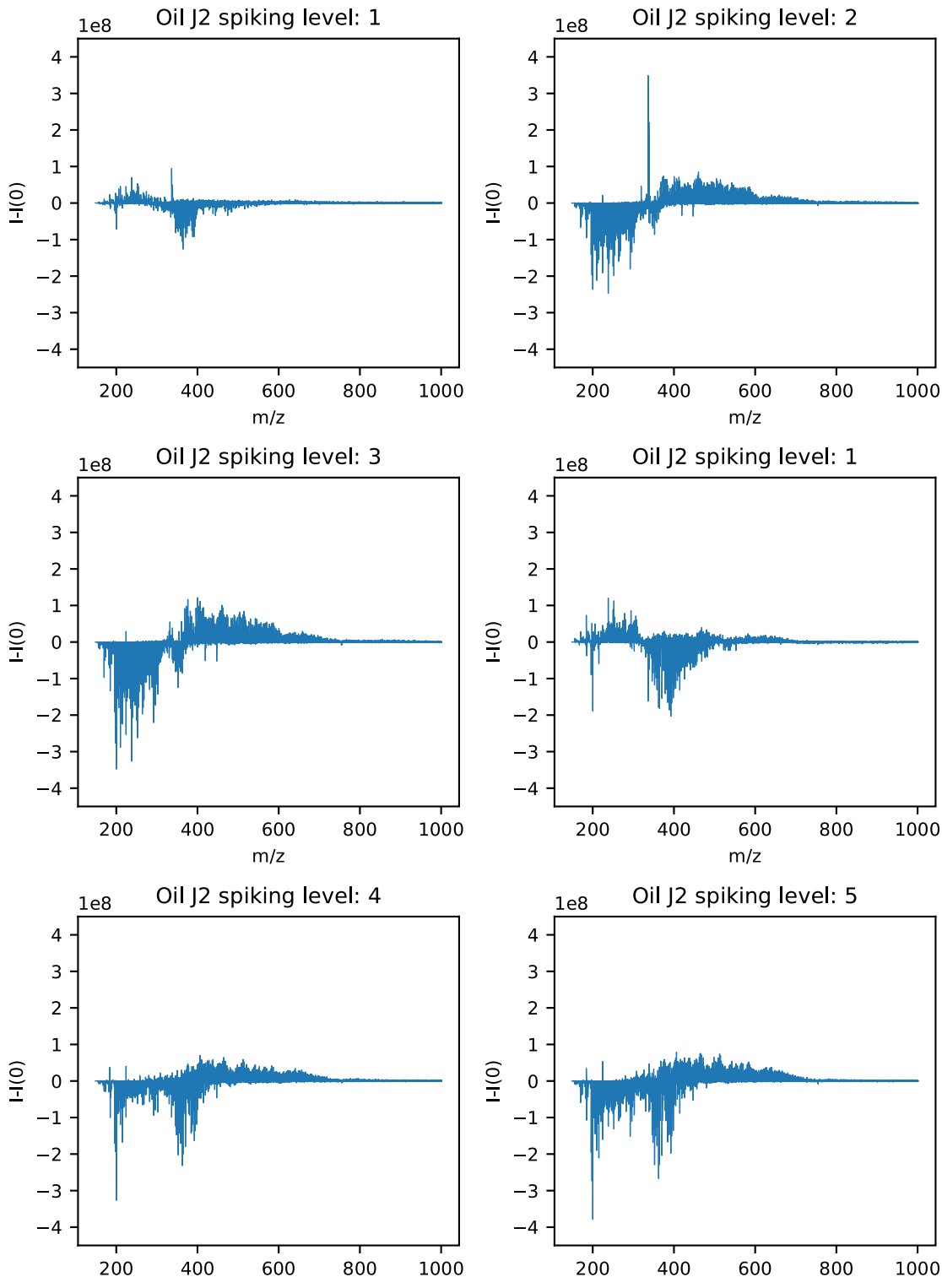

**Fig 5. Mass spectra of samples from the hydrate phases for oil J2 with spiking level 0 subtracted from each of the spiking levels 1–5.**

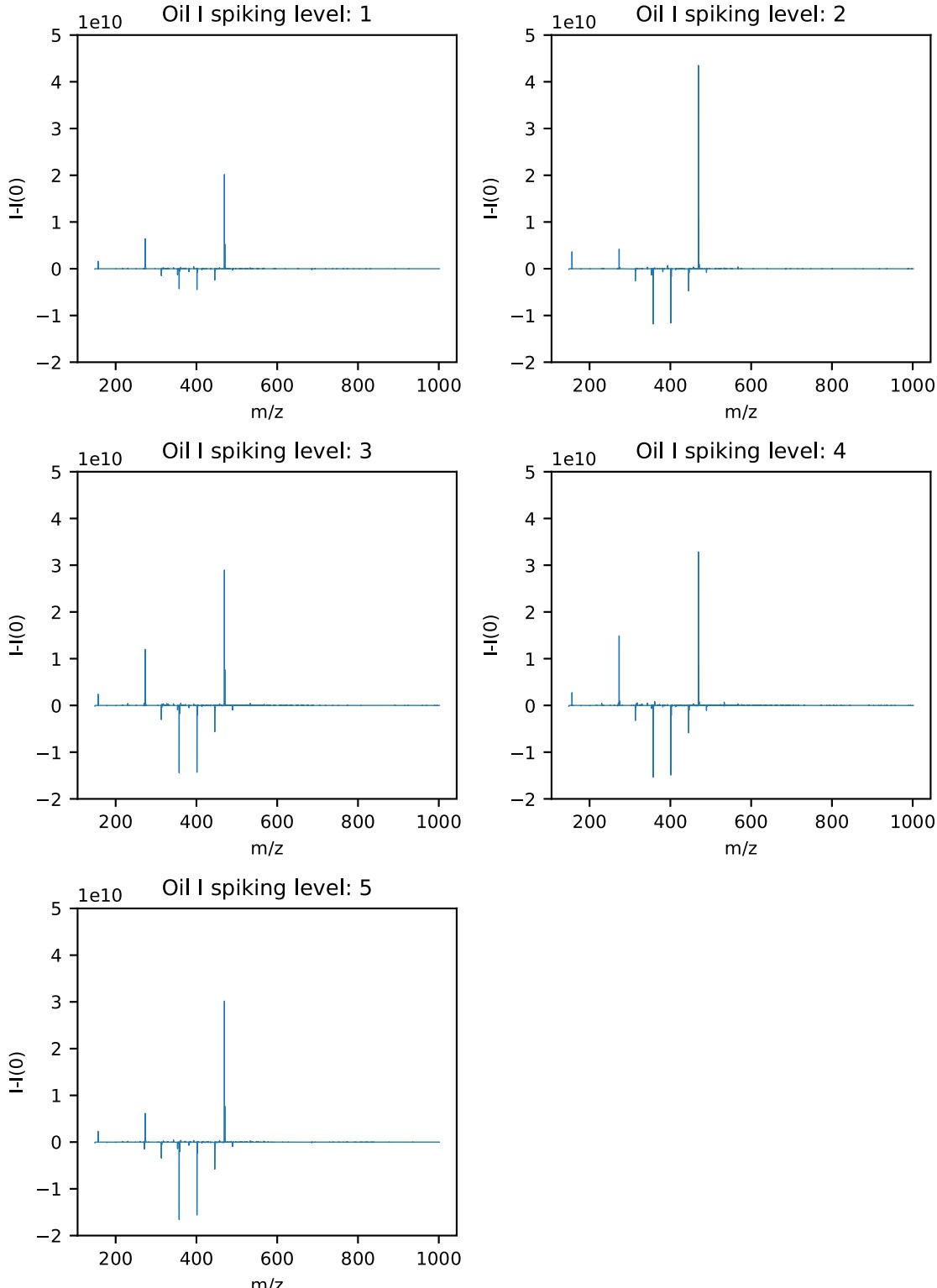

**Fig 6. Mass spectra of samples from the hydrate phase for oil I with spiking level 0 subtracted from each of the spiking levels 1–5.**

**Table 2. Peaks incresing for oil I.**

| m/z | Formula | DBE | H/C | Adduct | Molecular formula |
|---|---|---|---|---|---|
| 273.17 | $C_{12}H_{26}O_5$ | 0 | 2.17 | Na | 250.1780 |
| 469.32 | $C_{28}H_{46}O_4$ | 10 | 1.64 | Na | 446.3396 |

The m/z values with increasing trend as spiking levels increased for oil I and their molecular formula, DBE, H/C-ratio, which adduct the molecule has, Na or $H^+$, and the molecular weight.

## Discussion

The results from this work indicated that using machine learning-based variable selection, it is possible to identify components related to hydrate formation. Several methods were tested, and PLS-DA was determined as the best performing method with an accuracy of $0.62 \pm 0.12$ over 25 different training and test set splits. To determine a representative range, 25, 50, 75 and 100 training and test set splits were run. This sensitivity evaluation indicated that increasing the amount of splits above 25 would not affect the standard deviation significantly. Variable selection models can be prone to overfitting as they consume degrees of freedom, but when using an independent test set, overfitting of the models are counteracted. For each of the 25 times a new model was fitted, the variables selected as important by the model, based on their variable importance score, were extracted. The variables were extracted from the model with the highest accuracy score as that is the model that most accurately predicts the differences between the bulk samples and the hydrate samples, and therefore selects the variables with the highest probability of being related to hydrate formation.

From PLS-DA, 23 variables were selected as important by all of the 25 models and they were identified with their molecular formula, DBE and H/C-ratio. The variables selected ranged from m/z 271.19 to 763.61 and the carbon chains from $C_9$ to $C_{49}$. The DBE numbers show

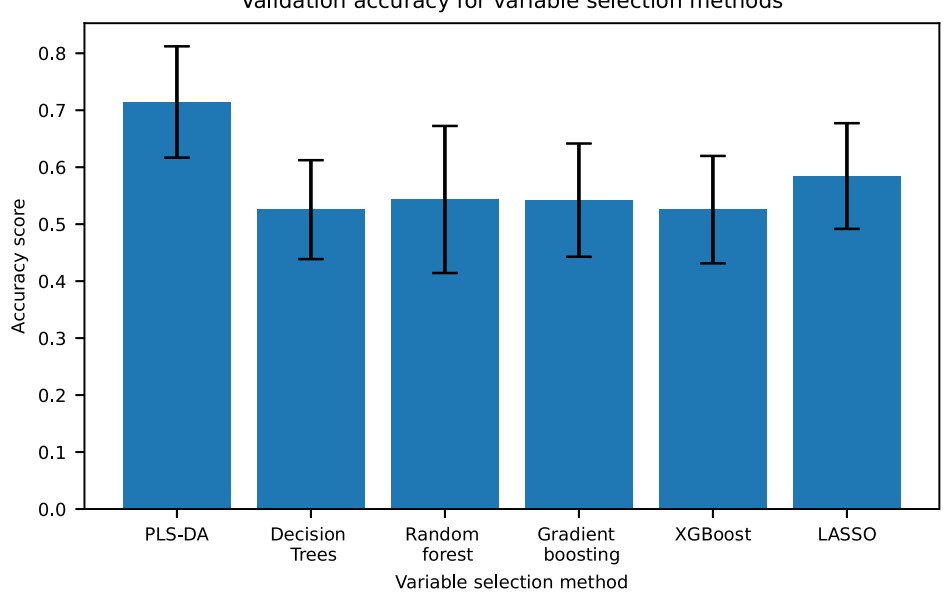

**Fig 7. Accuracy scores for the variable selection methods with error bars showing the standard deviation over 25 training/test set splits.**

**Table 3. Performance of the variable selection methods.**

| Method | Accuracy | Standard deviation | No. of variables selected | No. of variables selected in every model |
|---|---|---|---|---|
| PLS-DA | 0.62 | 0.12 | 132 | 26 |
| Decision Trees | 0.53 | 0.13 | 98 | 0 |
| Random Forest | 0.54 | 0.09 | 24929 | 44 |
| Gradient Boosting | 0.54 | 0.10 | 12364 | 0 |
| XGBoost | 0.53 | 0.09 | 786 | 0 |
| LASSO | 0.58 | 0.09 | 334 | 0 |

Performance for each variable selection method, their average accuracy, standard deviation, the number of variables selected during the 25 models and the number of variables that were selected in every model.

6 saturated variables, with DBE of 0, and the highest DBE was 10. The average weight of asphaltenes is ∼750 Da [52], and some of the selected m/z-values were in the range 705–763, indicating that these could be asphaltenes. The asphaltenes with hydrate inhibiting properties often have higher oxygen and sulfoxide content, higher acidity and lower DBEs [15, 53]. All of the possible asphaltenic structures exhibit these properties, and could thereby be related to the positive WI for oil J2 and I. Further studies on the oil samples by extracting and analysing the

**Table 4. The variables selected by PLS-DA through VIP.**

| m/z | Formula | DBE | H/C | Adduct | Molecular weight |
|---|---|---|---|---|---|
| 271.19 | $C_{13}H_{28}O_4$ | 5 | 2.15 | Na | 248.1988 |
| 273.17 | $C_{12}H_{26}O_5$ | 0 | 2.17 | Na | 250.1780 |
| 313.24 | $C_{17}H_{32}N_2O_3$ | 3 | 1.88 | $H^+$ | 312.2491 |
| 326.38 | $C_{22}H_{47}N$ | 0 | 2.14 | $H^+$ | 325.3709 |
| 353.27 | $C_{19}H_{38}O_4$ | 1 | 2.00 | Na | 330.2770 |
| 357.26 | $C_{18}H_{38}O_5$ | 0 | 2.11 | Na | 334.2719 |
| 359.24 | $C_{17}H_{36}O_6$ | 0 | 2.12 | Na | 336.2512 |
| 360.32 | $C_{22}H_{43}NO$ | 2 | 1.95 | Na | 337.3345 |
| 361.22 | $C_{16}H_{34}O_7$ | 0 | 2.13 | Na | 338.2305 |
| 381.30 | $C_{21}H_{42}O_4$ | 1 | 2 | Na | 358.3083 |
| 393.30 | $C_{22}H_{42}O_4$ | 2 | 1.91 | Na | 370.3083 |
| 397.18 | $C_{18}H_{30}O_8$ | 4 | 1.67 | Na | 374.1941 |
| 401.29 | $C_{26}H_{40}OS$ | 9 | 1.53 | $H^+$ | 400.2710 |
| 408.31 | $C_{22}H_{43}NO_4$ | 2 | 1.95 | Na | 385.3291 |
| 425.41 | $C_{26}H_{52}N_2O_2$ | 2 | 2.00 | $H^+$ | 424.4029 |
| 445.31 | $C_{22}H_{46}O_7$ | 0 | 2.09 | Na | 422.4029 |
| 451.19 | $C_{21}H_{29}O_9$ | 6 | 1.52 | Na | 328.2046 |
| 457.28 | $C_{22}H_{42}O_7$ | 2 | 1.91 | Na | 434.2880 |
| 469.31 | $C_{24}H_{46}O_7$ | 2 | 1.92 | Na | 446.3244 |
| 469.32 | $C_{28}H_{46}O_4$ | 10 | 1.64 | Na | 446.3396 |
| 705.58 | $C_{42}H_{82}O_4S$ | 4 | 1.95 | Na | 682.5934 |
| 750.52 | $C_{36}H_{81}N_3O_7SV$ | 2 | 2.25 | $H^+$ | 749.5157 |
| 763.61 | $C_{45}H_{80}N_4O_3$ | 8 | 1.78 | Na | 740.6180 |

Table of the 23 m/z values selected in every of the 25 PLS-DA models, their molecular formulas, DBE numbers, hydrate-carbon ratio, which adduct the molecule has, sodium or (Na) or hydrogen ($H^+$), and the molecular weight.

asphaltenes may confirm this. One variable, m/z 469.32, follows the general molecular formula for naphthenic acids. Other m/z values appear to have properties that could possibly define them as naphthenic acids, two or more oxygen molecules, DBEs indicating unsaturation and H/C-ratios below 2. As naphthenic acids are suggested to be related to hydrate active components [8], it is therefore likely that they contribute to the positive wetting index for oil J2 and I. Several of the selected variables have molecular formulas corresponding to $C_nH_{2n+2}$ and have a DBE of zero. They have carbon chains between $C_{12}$ and $C_{22}$ and contain either large amounts of oxygen ($O_5$ or more) or nitrogen. It is therefore probable that these are polyethylene glycol (PEG) molecules stemming from production chemicals used to treat flow assurance issues during extraction and processing of the crude oil [54].

By conducting the successive accumulation procedure for a given oil, generations with possibly increased concentration of hydrate active components could be accumulated. The oils with positive WI, likely to exhibit non-plugging properties, should thereby achieve an increase in the components related to anti-agglomeration, making their identification easier. The PCA scoreplots in Fig 2 show that the crude oils are distinguishable from the spiking samples in both the bulk phase and the hydrate phase. Additionally, the PCA scoreplots in Fig 3 show that the different spiking levels are separated, indicating that there were differences between the samples extracted from each spiking level. The spiking procedure therefore altered the composition of the oils. The variables selected by PLS-DA were also identified as increasing in the hydrate phase spiking fractions for oil A and I supporting the theory of accumulation.

The mass spectra for oil J2 in Fig 5 showed that no distinct m/z values increased as the spiking levels increased. However, for spiking level 2, 3, 4 and 5, the area between m/z 400 and 600 increased, indicating that the variables relevant for hydrate formation could lie in this m/z region. Another possible explanation could be that this oil is saturated with hydrate active components, and the spiking procedure therefore would not change the composition of the oil. This fits well with the WI of +0.44 for oil J2, indicating little or no plugging. It is therefore likely that oil J2 contains more hydrate active components than the oils with lower WI.

The results from the variable selection methods showed that the two linear methods, PLS-DA and LASSO, achieved higher accuracy scores than the tree-based methods. Linear methods are more robust and less susceptible to changes in the data. As there were variations in the accuracy for the models using different training and test set splits, the tree-based methods were likely affected negatively.

The molecular formulas presented in this paper are only suggestions of the most likely molecular formulas from the DataAnalysis software. As the mass of the molecule increases, the amount of possible structures and formulas also increases. Accordingly, the uncertainty of the suggested formulas increases with the mass of the molecule. Nonetheless, the structures give an indication of the nature of the molecules related to hydrate formation, and can be used to indicate whether they are i.e. asphaltenes, acids or alkanes.

For any complex data matrix, there are often assumptions that some of the data is noise and unrelated to the desired prediction. With the methodology presented in this paper, we show that it is possible to extract relevant information from complex data and relate it to the chemical composition of the samples. Thus, the proposed methods can be used in any application where there is a need for extracting, identifying and evaluating important variables.

## Further studies

When the m/z values of components related to hydrate formation are identified, the next step will be to determine the molecular structures with higher certainty. This can be done by isolation and fragmentation by FT-ICR MS, making it easier to identify the structures of

complicated molecules. When the compounds are found, they can be tested with the oils to evaluate how their presence changes the characteristics of the oils and the formation of hydrates.

## Conclusion

In this study, machine learning-based variable selection was used to identify components related to hydrate formation. A successive accumulation procedure was performed to increase the concentration of the hydrate active components. PCA demonstrated the difference between the spiking levels and the crude oils, establishing that the spiking procedure alters the sample composition significantly, suggesting that hydrate active components have been accumulated. Variable selection methods such as Decision Trees, Random Forest, Gradient Boosting, XGBoost, LASSO regularisation and PLS-DA through VIP were tested to identify the hydrate active components. The best performing prediction model was obtained using PLS-DA which gave an average accuracy of 0.62±0.12 over 25 different training and test set combinations. From the 25 models, 23 variables were selected as important in every model, and their molecular formulas were determined in an attempt to identify molecules related to hydrate formation. Some of the variables were identified as possible asphaltenic structures which could be related to the positive WI for the oils.

Identifying variables in the oil related to hydrate formation takes us one step closer to identifying the naturally occurring hydrate active components.

## Supporting information

**S1 Fig. Experimental set-up.** Picture of the autoclave used for the hydrate formation and spiking experiments. It consists of a sapphire cell between two titanium grad II flanges. Pressure, temperature and conductance is measured inside the sapphire cell. A motor is mounted above the cell driving a stirrer through a magnetic connection.
(PDF)

**S2 Fig. Determining the threshold for VIP Plotting of the VIP values for the 25 PLS-DA models with 20 components.** The curve flattens around 5 which was selected as the threshold.
(EPS)

**S3 Fig. Determining the optimal training/test set split.** Accuracy scores for the classification methods showing the mean accuracy over 25, 50, 75 and 100 training/test set splits and standard deviations.
(EPS)

## Author Contributions

**Conceptualization:** Elise Lunde Gjelsvik, Kristin Tøndel.

**Data curation:** Elise Lunde Gjelsvik.

**Formal analysis:** Elise Lunde Gjelsvik.

**Funding acquisition:** Martin Fossen, Kristin Tøndel.

**Methodology:** Elise Lunde Gjelsvik, Martin Fossen, Anders Brunsvik.

**Project administration:** Martin Fossen.

**Supervision:** Martin Fossen, Anders Brunsvik, Kristin Tøndel.

**Writing – original draft:** Elise Lunde Gjelsvik.

**Writing – review & editing:** Martin Fossen, Anders Brunsvik, Kristin Tøndel.

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
