## [Decision Letter · Decision Letter 0]

14 Jun 2022

PONE-D-22-13309Using machine learning-based variable selection to identify hydrate related components from FT-ICR MS spectraPLOS ONE

Dear Dr. Gjelsvik,

Thank you for submitting your manuscript to PLOS ONE. After careful consideration, we feel that it has merit but does not fully meet PLOS ONE’s publication criteria as it currently stands. Therefore, we invite you to submit a revised version of the manuscript that addresses the points raised during the review process.

We look forward to receiving your revised manuscript.

Kind regards,

Joseph Banoub, Ph,D., D. Sc.

Academic Editor

PLOS ONE

Journal Requirements:

Reviewers' comments:

Reviewer's Responses to Questions

**Comments to the Author**

1. Is the manuscript technically sound, and do the data support the conclusions?

Reviewer #1: Yes

Reviewer #2: Partly

2. Has the statistical analysis been performed appropriately and rigorously? 

Reviewer #1: Yes

Reviewer #2: I Don't Know

3. Have the authors made all data underlying the findings in their manuscript fully available?

Reviewer #1: Yes

Reviewer #2: No

4. Is the manuscript presented in an intelligible fashion and written in standard English?

Reviewer #1: Yes

Reviewer #2: Yes

5. Review Comments to the Author

Reviewer #1: I like this paper. It addresses an important problem, has clear logic wrt analysis, uses good experimental methods and instrumentation, is cross-cultural in its multivariate data modelling and has good data analysis and conclusion.

I have some minor concerns, listed in the enclosed Word document.

Reviewer #2: PONE-D-22-13309,

The paper from Gjelsvik et al. deals with the use of modern data mining technologies for the understanding of interaction of gas hydrates with crude oil samples using data from ultrahigh resolution FT-ICR mass spectrometry.

Overall, the topic is of interest, as especially in engineering the interactions between an oil and the surrounding is of interest for a seamless process. And FT-MS are the instruments of choice for such detailed and complex analytical problems. Still, the way the manuscript is written it is suffering from combining different methodologies and technologies but without really explaining them all. A lot of details that are necessary for the understanding of the research is missing or not described in an understandable way. Please keep in mind, that the journal is of multidisciplinary character, which needs attention.

Some detailed comment:

1. The experimental is not detailed enough about how the data were obtained from mass spectrometry. These data are the base of everything that follows and to see it the methods are solid the authors need to show that the data are solid. How were the data obtained, what has been done with them, are the transients used directly and which way was FT done (eFT or mFT?). The pictures in Fig. 4-6 look like not fully processed spectra, where the phasing may be missing. This is not described. How was the data depth for recording of the transients? For the analysis of a crude oil sample this is not really sufficient.

2. The data were obtained only using positive ion MS. But the data shown in the tables are all containing oxygen species. They probably would be better recorded in negative mode ESI. This is missing.

3. The experimental procedure is not really conclusive. The authors write that they were using an HPLC but did not name the column that was used.

4. Then the sample was fairly well diluted and of this diluted sample only 100 µl was injected. The values for infusion mode MS measurements of crude oil in the literature are from 100 to 500 ppm which are then infused over a certain time frame to generally generate at least from 50 to 100 spectra that are then averaged. Here, without having access to the original data this is not good enough to see if the method is actually working. And if the authors want to understand the interaction between the gas hydrates and oil they need to have solid data.

5. The molecular interaction between oil and gas hydrates are not well described. What is the main topic here, is it some type of hydrogen bonding that is proposed or other types of molecular interactions? But deposits of such different types of compouds can also be caused by differences in solubility. Does this play a role? It is not discussed.

6. Some more engineering type of terms have not been defined. What is for example the wetting index and were is it good for? Explain briefly because not everybody will understand this. Same for other more engineering terms.

7. The data in the tables are not correct. C27H38N4O3 does not have an m/z of 156,44 but rather 466.2943909 for the non-charged molecule. But it is an ion. So all data in the tables are not correct, as the formulas are not correlating with the m/z values.

8. How certain are the authors about the molecular compositions and their validity. Some are very much debatable, like C13H30V. How can a vanadium bind with a pure hydrocarbon. This goes into coordination chemistry and here are no free electron pairs to make a complex bond. Unlikely. Others like the …N4V… containing formulas could correlate with vanadium porphyrins that were already reported in the literature, but the formula are not really fitting here either. So this needs to be better interpreted how these formulas really come together and if they are reasonable.

9. Figures 4-6 are way too small

There are a few minor corrections but at this point I will not go into this because before the manuscript is ready for publication there is still quite some work to be done to show that it is worth it.

6. PLOS authors have the option to publish the peer review history of their article (what does this mean?). If published, this will include your full peer review and any attached files.

Reviewer #1: **Yes: **Harald Martens

Reviewer #2: No

---

## [Author Response · Author response to Decision Letter 0]

18 Jul 2022

Additionally to the changes proposed by the reviewers, changes has been made to the text to fit the PLOS ONE's style requirements. See all changes in the manuscript version with track changes. 

The financial disclosure stated in the paper is the correct one, and it reads: “The authors thank the Norwegian Research Council, Equinor ASA, OMV (Norge) AS, Wintershall DEA Norge AS and TotalEnergies for funding. This work is a part of the Knowledge-Building Project for Industry (PETROMAKS 2), Project number: 294636 “New Hydrate Management: New understanding of hydrate phenomena in oil systems to enable safe operation within the hydrate zone”.

---

## [Editor Report · Decision Letter 1]

3 Aug 2022

Using machine learning-based variable selection to identify hydrate related components from FT-ICR MS spectra

PONE-D-22-13309R1

Dear Dr. Gjelsvik,

We’re pleased to inform you that your manuscript has been judged scientifically suitable for publication and will be formally accepted for publication once it meets all outstanding technical requirements.

Kind regards,

Joseph Banoub, Ph,D., D. Sc.

Academic Editor

PLOS ONE

Additional Editor Comments (optional):

The authors have answered all demanded queries of the referees.
---

## [Editor Report · Acceptance letter]

8 Aug 2022

PONE-D-22-13309R1 

Using machine learning-based variable selection to identify hydrate related components from FT-ICR MS spectra 

Dear Dr. Gjelsvik:

I'm pleased to inform you that your manuscript has been deemed suitable for publication in PLOS ONE. Congratulations! Your manuscript is now with our production department. 

Kind regards, 

on behalf of

Dr. Joseph Banoub 

Academic Editor

PLOS ONE